# Factors Associated with Hepatitis B Medication Adherence and Persistence among Underserved Chinese and Vietnamese Americans

**DOI:** 10.3390/jcm11030870

**Published:** 2022-02-07

**Authors:** Aisha Bhimla, Lin Zhu, Wenyue Lu, Sarit Golub, Chibuzo Enemchukwu, Elizabeth Handorf, Yin Tan, Ming-Chin Yeh, Minhhuyen T. Nguyen, Min Qi Wang, Grace X. Ma

**Affiliations:** 1Center for Asian Health, Lewis Katz School of Medicine, Temple University, 3440 N. Broad St., Philadelphia, PA 19140, USA; aisha.bhimla@temple.edu (A.B.); lin.zhu@temple.edu (L.Z.); wenyue.lu@temple.edu (W.L.); ytan@temple.edu (Y.T.); 2Department of Urban Health and Population Science, Lewis Katz School of Medicine, Temple University, 3440 N. Broad St., Philadelphia, PA 19140, USA; 3Department of Psychology, Hunter College, City University of New York, 695 Park Ave., New York, NY 10065, USA; sgolub@hunter.cuny.edu (S.G.); ce468@hunter.cuny.edu (C.E.); 4Cancer Prevention and Control Program, Fox Chase Cancer Center, Temple University Health System, 3401 N. Broad St., Philadelphia, PA 19140, USA; elizabeth.handorf@temple.edu; 5Nutrition Program, Hunter College, City University of New York, 695 Park Ave., New York, NY 10065, USA; myeh@hunter.cuny.edu; 6Department of Medicine, Section of Gastroenterology, Fox Chase Cancer Center, Temple University Health System, 3401 N. Broad St., Philadelphia, PA 19140, USA; Minhhuyen.Nguyen@fccc.edu; 7School of Public Health, University of Maryland, 4200 Valley Dr., College Park, MD 20742, USA; mqw@umd.edu

**Keywords:** hepatitis B, Asian Americans, medication adherence, information–motivation–behavioral skills model

## Abstract

Background: Hepatitis B virus (HBV) infection disproportionately affects Asian Americans in the United States, while this population faces low adherence to HBV treatment. Using the information–motivation–behavioral skills model (IMB), the study aims to examine medication adherence and persistence among Chinese and Vietnamese people with HBV. Methodology: Study participants were recruited between March 2019 and March 2020 and were enrolled through multiple recruitment approaches in the Greater Philadelphia Area and New York City. The study is an assessment of the baseline data on medication adherence, HBV-related knowledge, motivation of HBV medication treatment, self-efficacy about HBV medication treatment, and socioeconomic status. Results: Among 165 participants, 77.6% were Chinese and 22.4% were Vietnamese Americans. HBV-related knowledge/information, motivation, and self-efficacy were all positively associated with having medium/high medication adherence. Multilevel mixed-effects generalized linear regression revealed that living more than 10 years in the U.S. (OR = 4.24; *p* = 0.028) and greater information–knowledge about HBV (OR = 1.46; *p* = 0.004) were statistically associated with higher odds of medium/high medication adherence. Moreover, greater HBV-related knowledge/information (OR = 1.49; *p* = 0.023) and greater motivation towards HBV treatment adherence (OR = 1.10; *p* = 0.036) were both associated with a higher likelihood of medication persistence. Conclusion: Our findings provided significant implications in designing behavioral interventions focused on self-efficacy, information, and motivation to promote better medication adherence among Asian Americans living with HBV.

## 1. Introduction

Cancer is the leading cause of morbidity and mortality among Asian Americans [1,2]. Liver cancer incidence and mortality are significantly high, disproportionally affecting Asian residing in the United States (U.S.). The incidence of liver cancer for Asian American men is 19.9 per 100,000 and women is 7.4 per 100,000, in comparison to 10.8 per 100,000 for non-Hispanic white men and 3.7 for non-Hispanic white women, respectively [2,3]. Chronic Hepatitis B Virus (HBV) disproportionately affects Asian Americans, where they represent 6% of the total U.S. population but make up 58% of HBV-linked hepatocellular carcinoma (HCC) cases [4]. HBV is the leading risk factor for developing HCC, which has been known to create a >100-fold increased risk for developing HCC [5]. More specifically, Vietnamese and Chinese Americans suffer from disproportionate rates of HBV infection, as they represent countries with endemic HBV infection at the highest rates in Asia [6,7,8]. Community-based studies within the United States have illustrated infection rates in these populations as high as 52% [8].

A major problem faced among this critical population is low adherence to HBV treatment [8,9]. Community-based studies have illustrated that Asian Americans infected with HBV are severely untreated, with one study showing very low levels of adherence to antiviral therapy [9]. In this large-scale retrospective cohort study, only 50% of treatment-eligible patients were receiving antiviral therapy within 12 months of care [9]. This adherence rate is lower compared to two cohort studies conducted in the United States, where there were adherence rates of 88% and 98% [10,11]. There are several factors that contribute to non-adherence to medication, stemming from psychosocial, environmental, and health system influences [12,13]. Furthermore, adherence may be affected by adverse reactions to the medication; however, antivirals for chronic HBV are known to have less side effects, and adverse outcomes are rare [14]. Fewer studies have been conducted to understand the factors that affect health outcomes and behaviors among Asian American HBV infected patients. A systematic review of studies on factors that affect medication adherence to nucleos(t)ide analogues for HBV across all patient populations were younger age and psychiatric diagnosis as the strongest predictors of non-adherence [13]. In addition to these factors, Allard and colleagues (2017) found that the language barriers between clinicians and patients may be a predictor of low adherence to antiviral therapy. Furthermore, the chronic nature of HBV can have significant impacts on quality of life and mental health throughout the lifespan. More research is needed regarding understanding what sociodemographic and psychosocial factors predict adherence to medication among HBV Asian American patients.

The consequences of non-medication adherence among those infected are increased virological failure and a higher risk of disease progression to chronic HBV leading to conditions such as cirrhosis as well as HCC [13]. Understanding what factors contribute to medication adherence and persistence can be crucial for providing appropriate support for maintaining long-term treatment adherence and sustaining a virological response through interventions and strategies for high-risk populations affected by HBV.

The information–motivation–behavioral skills model (IMB) is commonly adopted to understand and support self-management and adherence among patients living with chronic conditions [15,16,17]. According to this model, patients’ health behavior results from three core components: accurate, pertinent, and high-quality information about an illness and treatment; personal and social motivation for engaging in a health behavior; and adequate skills and self-efficacy to perform the health behavior [15]. Personal motivation is comprised of individual’s attitudes and beliefs towards performing the behavior [17]. Two studies using the IMB model have been applied to patients diagnosed with cervical and endometrial cancer with regards to adherence to engaging in self-care related behaviors [15]. Similarly, research on hepatitis C virus (HCV) treatment adherence has indicated that information, as well as personal and social motivation, were key determinants for medication adherence to HCV antiviral medication, and gaps in knowledge can lead to HCV reinfection [18]. The IMB model is a useful theoretical framework for understanding medication adherence and persistence among HBV patients, but has not been previously applied to understudied HBV positive Asian American ethnic populations.

Thus, this study aims to examine the sociodemographic and IMB-related psychosocial factors, specifically knowledge–information, motivation, and behavioral skills, on self-efficacy as predictors of low versus medium/high medication adherence and medication persistence among a sample of Chinese and Vietnamese HBV positive patients.

## 2. Materials and Methods

### 2.1. Study Population and Data Collection

For this current study, we examined associations between information, motivation, and behavioral skills (IMB model) constructs and HBV medication adherence and persistence among Chinese and Vietnamese American HBV positive patients. This article focuses on the baseline data analysis of a randomized controlled intervention trial to increase adherence to HBV follow up monitoring and medication adherence. A total of 382 eligible participants were recruited from clinical settings and community-based organizations (CBO) in the greater Philadelphia and New York City areas. Participants were eligible to participate in the study if they: (1) were aged 18 and above; (2) self-identified as Chinese or Vietnamese ethnicity; (3) had chronic HBV infection with positive HBV surface antigen (HBsAg); (4) were diagnosed for over 12 months; (5) were non-compliant to HBV monitoring and treatment guidelines for more than six months; and (6) had cell phone and internet access.

The study was approved by the Western Institutional Review Board (WIRB). Participants recruited from clinical settings were identified through medical record review by authorized staff from collaborating health clinics, pharmacies, or community health centers. Community-based organizations included racial/ethnic based educational and community centers, religion-based organizations, and senior centers. An invitation letter with study information was sent from clinical partners to eligible participants. A follow-up phone call was also made to those eligible participants from the respective clinical center. Eligible participants who were interested in participating in the study contacted the study team and were enrolled. Participants from non-clinical sites (e.g., CBOs) were screened for eligibility by study team staff, and chronic hepatitis B status was confirmed using notes from doctors’ offices or medical diagnosis reports provided by participants. Some of these eligible participants were enrolled in a previous intervention study. After participants completed written informed consent, they completed a baseline assessment via either in-person self-administration or phone call in their preferred language (e.g., English, Chinese, or Vietnamese languages).

The data reported in this study represent analysis of baseline assessments, i.e., prior to initiation of the intervention. Baseline assessment included measures of demographics, HBV infection history, HBV monitoring behavior, medication adherence and persistence, treatment motivation and self-efficacy, knowledge about HBV prevention, diagnosis and treatment, and mental health status. It took approximately 30 min to complete the baseline assessment. Since the current analysis focused on predictors of medication adherence at baseline, we excluded participants who reported that they were not currently receiving HBV medication treatment (n = 217). This left us with an analytic sample of 165 participants (43%) for this analysis.

### 2.2. Measures

#### 2.2.1. Outcome Measures

The first outcome measure was HBV medication adherence, assessed with the Morisky 8-Item Medication Adherence Scale (MMAS-8) [19]. The MMAS-8 is a valid and reliable instrument that examines adherence to prescribed medication by asking eight questions, with seven questions of options yes/no and the eighth question having five response options. In a systematic review of studies, the MMAS-8 had strong psychometric properties, including strong reliability and validity for measuring medication adherence among patients living with chronic conditions [19]. Consistent with past research, the study outcome was categorized as a binary indicator, where low adherence was defined as a score of 0–5 and medium/high adherence was designated as a score of 6 to 8 on the MMAS-8, i.e., good adherence [20].

The second outcome measure examined in this study was HBV medication persistence, measured with the question: “Since your last doctor’s visit, was there a time when you stopped taking your HBV medication for any reason?” The answers were coded into two categories. Those who reported that they never stopped taking their medication were categorized as “persistent”, and others as “not persistent.”

#### 2.2.2. IMB Constructs

Information (HBV-related knowledge) was examined with a 10-item scale that had been validated in Asian Americans with chronic HBV in our previous study [21]. Specifically, participants answered “False”, “True”, or “Don’t know” to 10 HBV knowledge statements such as “People will feel sick if they are infected with hepatitis B” and “Regular monitoring and treatment can reduce liver damage caused by chronic hepatitis B”. One point was assigned to correct answers and zero points to wrong answers. We computed the accumulated HBV knowledge score by summing the points from all 10 items. The final knowledge score ranged from 0 to 10, with a higher numeric value indicating a higher level of HBV-related knowledge.

Motivation for chronic HBV management was accessed with 10 items on a five-point Likert scale (from 1 “strongly agree” to 5 “strongly disagree”). Examples of the items include “I get frustrated taking my HBV medications because they remind me that I am HBV+” and “My healthcare provider doesn’t give me enough support when it comes to taking my medications as prescribed.” The total motivation score was the summation of the responses to the 10 items. The score ranged from 5 to 50, with a higher numeric value indicating a higher level of motivation related to CHB management

Behavior (self-efficacy for chronic HBV management) was measured with 13 items asking participants how confident they were in taking HBV medications under various situations. The answers ranged from 0 “not at all” to 3 “extremely sure”, which were summed up to compute the final self-efficacy score. The self-efficacy score ranged from 0 to 26, with a higher numeric value indicating greater confidence in taking HBV medications as doctors guided.

#### 2.2.3. Socioeconomic Factors

Participants’ age in years, gender, ethnicity, birthplace, years living in the U.S., marital status, health insurance coverage, having a regular physician, and English proficiency were included as predictors of HBV medication adherence and persistence outcomes.

### 2.3. Statistical Analysis

We conducted chi-square test and t-test to examine the associations between sociodemographic/psychosocial factors and the outcomes: medication adherence and persistence. We also fitted multilevel mixed-effects generalized linear models (GLMs) to identify the significant predictors of the outcomes while accounting for the sample clustering by recruitment site, state, and those who were recruited from our previous intervention study. To avoid model over-fitting, we pre-specified demographic factors of primary interest; these variables were included in the multilevel mixed-effects GMLs. All data analyses were conducted in Stata 16 [22]. A *p* value that is smaller than 0.05 was considered statistically significant.

## 3. Results

Table 1 presents the descriptive and bivariate analysis results. Among 165 patients, 43.6% (n = 72) were female and 56.4% (n = 93) were male, with 77.6% being of Chinese and 22.4% Vietnamese ethnicity. The average age of patients was 52.8. The vast majority of participants reported being married (85.4%). Nearly all patients reported having insurance (90.9%) and having a physician to visit regularly (94.9%). Nearly all patients were born outside of the United States (98.8%) and most participants had been residing in the United States for 10 years or longer (87.3%). About three-quarters of the participants (73.9%) reported low English proficiency (not at all/not well). Mean HBV-related information–knowledge score, HBV motivation score, and self-efficacy scores were 5.4, 28.2, and 5.9, respectively. About two-thirds of the sample (66%, n = 109) reported medium/high rates of medication adherence, and 34% (n = 56) reported low medication adherence. In terms of medication persistence, 113 out of 155 participants with valid data on persistence (72.90%) reported that they were not persistently taking their antihypertensive medication since their last doctor’s visit.

The three IMB model constructs were positively associated with each other. Specifically, information–knowledge score was positively associated with motivation score (Pearson’s r = 0.14, *p* = 0.006) and behavioral skill–self-efficacy score (Pearson’s r = 0.16, *p* = 0.003). Motivation score was also positively associated with behavioral skill–self-efficacy score (Pearson’s r = 0.40, *p* < 0.001) (results not shown).

Bivariate analysis result indicated that those who were Vietnamese (vs. Chinese), had lived in the U.S. for 10 or more years (vs. <10 years), and were currently married (vs. not) were more likely to report medium/high medication adherence. In addition, all three constructs of the IMB model, i.e., knowledge–information, motivation, and self-efficacy, were all positively associated with having medium/high medication adherence (Table 1).

Bivariate analysis results indicated that those who were Vietnamese (vs. Chinese), those born outside the U.S. (vs. U.S.-born), and were currently married (vs. not) were more likely to report medication persistence. In addition, all three constructs of the IMB model, i.e., knowledge–information, motivation, and self-efficacy were all positively associated with medication persistence (Table 2).

Multilevel mixed-effects generalized linear regression on medical adherence (Table 3) revealed that living more than 10 years in the U.S. was statistically associated with higher odds of medium/high medication adherence (OR = 4.24; 95% CI: 1.17, 15.44; *p* = 0.028). With regards to the IMB model psychosocial constructs, greater information–knowledge about HBV diagnosis, prevention, and treatment was associated with greater odds of medication adherence (OR = 1.46; 95% CI: 1.12, 1.83, *p* = 0.004). Motivation for HBV treatment and self-efficacy were no longer statistically significant in the multivariable model.

Multilevel mixed-effects generalized linear regression on medical persistence (Table 4) revealed that greater HBV-related knowledge–information (OR = 1.49; 95% CI: 1.06, 2.10, *p* = 0.023) and greater motivation towards HBV treatment adherence (OR = 1.10; 95% CI: 1.01, 1.20, *p* = 0.036) were both associated with a higher likelihood of medication persistence, while self-efficacy was not a significant predictor (Table 4).

## 4. Discussion

This paper examined a set of theoretical predictors drawn from the IMB-model of health behavior in order to examine correlates of high self-reported medication adherence and persistence in a sample of Chinese and Vietnamese American patients living with HBV. Overall, a relatively high percentage of participants (66%) reported medium rates of medication adherence when measured by the MMAS-8. There has been considerable variability in medication adherence rates across patient populations in previous studies [13], and our analysis is the first, to our knowledge, to report rates specifically for this population. Rates of adherence may be slightly overestimated in our sample due to our recruitment methods; participants were recruited from clinical and community-based settings where they had access to some HBV-related support, and about 63% of participants were recruited from a previous HBV management intervention trial.

In our bivariate analyses, all three constructs of the IMB model—information, motivation, and behavioral efficacy—were positively associated with reports of medium/high medication adherence in this sample. These data suggest the importance of IMB constructs in better understanding and support HBV medication adherence in this population. In a multi-variable model, only information scores retained statistical significance. In a multi-variable model predicting medication persistence, both information and motivation were significantly positively associated with the outcome. In past applications of the IMB model, information is often the strongest predictor [23,24], followed by motivation. Past analyses have also used structural equations modeling to test the impact of information and motivation through behavioral skills [25], which was not possible given the sample size for this analysis.

Lower health literacy has been identified as a barrier to IMB constructs [23]; in this sample, English proficiency was not associated with adherence, but years in the United States was, suggesting that health literacy may be an indirect factor. The fact that motivation was a stronger predictor of persistence as opposed to adherence suggests that a linear motivation process for patients, in which information may be more important for behavior initiation than motivation, might be increasingly necessary for behavioral sustainment.

It is important to note that there is a large gap between high rates of health insurance and having a regular physician and adherence rates in this sample. Further research is necessary to examine the role of IMB factors for patients whose access to medication may be more hampered by other types of access. When patients are required to navigate additional barriers to attain medication, motivation and behavioral efficacy may become increasingly relevant.

Our data are subject to several limitations. Most importantly, the majority of our sample was recruited from a previous HBV medication adherence trial, and so may not be representative of HBV patients more generally, given that adherence rates were higher among our sample. These data are cross-sectional in nature, and so any ability to make causal inferences is limited.

Nonetheless, these data are among the first to examine the utility of IMB constructs in understanding and supporting medication adherence and persistence among Chinese and Vietnamese American patients living with HBV infection. Our findings suggest that intervening on and supporting IMB constructs may be useful in increasing both adherence and persistence in this critical high-risk population. Potentially feasible intervention components may include cultural tailoring for patient education, medication management, patient navigation, and psychological therapies (e.g., cognitive behavioral therapy) to promote medication and adherence.

## Figures and Tables

**Table 1 jcm-11-00870-t001:** Descriptive and bivariate analysis of sociodemographic predictors and psychosocial factors by HBV medication adherence status among Chinese and Vietnamese American HBV Patients (n = 165).

Sociodemographic Characteristics	Medication Adherence Category			
n (%)	Low (n = 56)	Medium/High (n = 109)	Total	*p*-Value
Age [m (SD)]	51.63 (13.19)	53.40 (13.13)	52.76 (13.14)	0.429
Gender				0.885
Female	24 (42.86)	48 (44.04)	72 (43.64)	
Male	32 (57.14)	61 (55.96)	93 (56.36)	
Ethnicity				<0.001
Chinese	55 (98.21)	73 (66.97)	128 (77.58)	
Vietnamese	1 (1.79)	36 (33.03)	37 (22.42)	
Born in the United States				0.629
Yes	1 (1.79)	1 (0.92)	2 (1.21)	
No	55 (98.21)	108 (99.08)	163 (98.79)	
Years living in the United States				0.045
Less than 10 years	11 (20.00)	9 (8.82)	20 (12.74)	
10 years or greater	44 (80.00)	93 (91.18)	137 (87.26)	
Marital Status				0.025
Married	43 (76.79)	97 (89.81)	140 (85.37)	
Other (Never married/divorced/widowed)	13 (23.21)	11 (10.19)	24 (14.63)	
Health insurance				0.959
No	5 (8.93)	10 (9.17)	15 (9.09)	
Yes	51 (91.07)	99 (90.83)	150 (90.91)	
Have a regular physician				0.298
No	4 (7.69)	4 (3.81)	8 (5.10)	
Yes	48 (92.31)	101 (96.19)	149 (94.90)	
English proficiency				0.202
Not at all/not well	38 (67.86)	84 (77.06)	122 (73.94)	
Well/very well	18 (32.14)	25 (22.94)	43 (26.06)	
IMB Model Psychosocial Factors	m (SD)	m (SD)	m (SD)	*p*-value
Information (knowledge score)	4.09 (2.64)	6.14 (1.92)	5.43 (2.40)	<0.001
Motivation for HBV treatment	25.34 (7.24)	29.76 (7.20)	28.22 (7.49)	<0.001
Behavioral skills (self-efficacy)	3.46 (4.80)	7.24 (8.07)	5.92 (7.31)	0.002

**Table 2 jcm-11-00870-t002:** Descriptive and bivariate analysis of sociodemographic predictors and psychosocial factors by HBV medication persistence status among Chinese and Vietnamese American HBV Patients (n = 155).

Sociodemographic Characteristics	Medication Persistence Category			
n (%)	Not Persistent (n = 56)	Persistent (n = 109)	Total	*p*-Value
Age [m (SD)]	50.76 (12.76)	54.03 (13.18)	53.15 (13.10)	0.168
Gender				0.088
Female	14 (33.33)	55 (48.67)	69 (44.52)	
Male	28 (66.67)	58 (51.33)	86 (55.48)	
Ethnicity				<0.001
Chinese	42 (100.00)	77 (68.14)	119 (76.77)	
Vietnamese	0 (0.00)	36 (31.86)	36 (23.23)	
Born in the United States				0.020
Yes	2 (4.76)	0 (0.00)	2 (1.29)	
No	40 (95.24)	113 (100.00)	153 (98.71)	
Years living in the United States				0.593
Less than 10 years	6 (15.38)	13 (12.04)	19 (12.93)	
10 years or greater	33 (84.62)	95 (87.96)	128 (87.07)	
Marital Status				0.005
Married	31 (73.81)	102 (91.97)	133 (86.36)	
Other (Never married/divorced/widowed)	11 (26.19)	10 (8.93)	21 (13.64)	
Health insurance				0.567
No	5 (11.90)	10 (8.85)	15 (9.68)	
Yes	37 (88.10)	103 (91.15)	140 (90.32)	
Have a regular physician				0.900
No	2 (5.13)	5 (4.63)	7 (4.76)	
Yes	37 (94.87)	103 (95.37)	140 (95.24)	
English proficiency				0.680
Not at all/not well	31 (73.81)	87 (76.99)	118 (76.13)	
Well/very well	11 (26.19)	26 (23.01)	37 (23.87)	
IMB Model Psychosocial Factors	m (SD)	m (SD)	m (SD)	*p*-value
Information/Knowledge score	3.53 (2.51)	6.09 (1.94)	5.41 (2.38)	<0.001
Motivation for HBV treatment	24.13 (6.72)	29.71 (7.33)	28.24 (7.57)	<0.001
Behavioral Skills/Self-Efficacy	2.78 (4.37)	7.37 (7.88)	6.16 (7.40)	<0.001

**Table 3 jcm-11-00870-t003:** Multilevel mixed-effects generalized linear regression of demographic and psychosocial predictors associated with HBV Medication Adherence.

Predictor	OR (95% CI)	*p*-Value
Female gender (ref. male)	0.94 (0.37, 2.41)	0.901
Currently married (ref. other)	0.53 (1.67, 1.70)	0.287
Lived in the U.S. 10 years or greater (ref. < 10 years)	4.24 (1.17, 15.44)	0.028
Have health insurance (ref. no)	0.55 (0.07, 4.42)	0.573
Have a regular physician (ref. no)	2.51 (0.26, 24.31)	0.426
Speaking English well/very well (ref. not at all/not well)	0.40 (0.13, 1.21)	0.103
Information (knowledge score)	1.46 (1.12, 1.83)	0.004
Motivation for HBV treatment	1.03 (0.96, 1.10)	0.406
Behavioral skills (self-efficacy)	1.02 (0.94, 1.10)	0.611

Abbreviations: OR = odds ratio; CI = confidence interval.

**Table 4 jcm-11-00870-t004:** Multilevel mixed-effects generalized linear regression of demographic and psychosocial predictors associated with HBV Medication Persistence.

Predictor	Odds Ratio (95% CI)	*p*-Value
Female gender (ref. male)	2.00 (0.68, 5.89)	0.206
Currently married (ref. other)	0.31 (0.08, 1.18)	0.087
Lived in the U.S. >= 10 years (<10 years)	1.10 (0.25, 4.83)	0.900
Have health insurance (ref. no)	1.72 (0.15, 19.24)	0.660
Have a regular physician (ref. no)	0.21 (0.01, 5.49)	0.352
Speaking English well/very well (ref. not at all/not well)	0.69 (0.16, 2.94)	0.612
Information (knowledge score)	1.49 (1.06, 2.10)	0.023
Motivation for HBV treatment	1.10 (1.01, 1.20)	0.036
Behavioral skills (self-efficacy)	1.01 (0.91, 1.11)	0.911

Abbreviations: OR = odds ratio; CI = confidence interval.

## Data Availability

The data presented in this study are available on request from the corresponding author.

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
