# Peer review of "Factors Associated with Hepatitis B Medication Adherence and Persistence among Underserved Chinese and Vietnamese Americans"

_jcm, 2022, doi:10.3390/jcm11030870_

Round 1

Reviewer 1 Report

Report on

Factors Associated with Hepatitis B Medication Adherence and Persistence Among Underserved Chinese and Vietnamese Americans

The main aim of this study is to examine medication adherence and persistence among Chinese and Vietnamese with HBV using the information-motivation-behavioral skills model (IMB). In my opinion,  the manuscript is well written; the methodology is reasonable and reliable; and the obtained results have many important implications.

However, I have a minor comment:

The study has only used baseline information of a randomized, controlled intervention trial  to arrive at their conclusion. I think this is not enough to give the conclusion.  The results may be biased by the person's feeling at the time they provide the information. As stated by the authors in the introduction section: "Most importantly, the majority of our sample was recruited from a previous HBV medication adherence trial." This may be a strong limitation that has already biased the study since the study was conducted to examine the associations between sociodemographic/psychosocial factors and the outcomes: medication adherence and persistence.

Author Response

We have attached our point-by-point response.

Reviewer 2 Report

Firstly I would like to inform that I don´t have any potential conflict of interest neither any other ethical concerns with regards to the paper:

Factors Associated with Hepatitis B Medication Adherence and Persistence Among Underserved Chinese and Vietnamese Americans

In this manuscript authors aimed to research which factors influence the HBV medication adherence in Asiatic population living in the U.S.A through the information–motivation–behavioral skills (IMB) model which in my opinion is correct for this purpose.

However, the interest for the readers is -in my view- low as it focuses in very specific subpopulations (Asians with HBC, living in U.S.A…), there are some important limitations in the recruitment of the participants -the paper looks like a rehash of previous and future research - but the authors admitted it and it shouldn't be an insurmountable inconvenience. The major flaw I found is that as far as I am concerned many people, especially in low- and middle-income countries like China or Vietnam (where HBV is furthermore endemic), do not know that they are infected, so they cannot access the appropriate treatments and this is the real problem; I did not know that adherence to these treatments was lower than that of any other pathology (or other population group) so I have asked the authors to emphasize this issue by providing some bibliography. I have also asked the authors to briefly explain what the treatment consists of (route of administration, duration, main adverse effects ...). To conclude the results of this type of research are not usually groundbreaking so the authors have to do their best in the discussion and to provide their expert insight into the problem advancing specific strategies by improving adherence (I have asked the authors to review it and to add new ideas).

Other than that, it was a pleasure to review this paper as the authors have shown an expertise and thoroughness that should be appreciated and methodology and results were presented in a clearly written and well-organized way. The information provided is comprehensive and I like the way it is shown.

I understand the article and although I am not a native English speaker it seems to be no language problems with it. I have superficially checked the paper for plagiarism by the and the low returned percentage of similarity would probably indicate that plagiarism has not occurred.

That is why my overall Recommendation is to reconsider after major revision.

I have also found several minor mistakes that should be corrected:

LINE 23: “while this population had low access and adherence to HBV treatment”, it is not the same to have low access than adherence so, please, provide quotations and explain this idea better.

LINE 37: change the conclusion: “our findings -explain briefly- provided significant implications in designing behavioral interventions -don't hold back, tell us at least one- to promote better medication adherence among Asian Americans living with HBV”.

LINE 57: provide a quotation for this “A major problem faced among this critical population is low access and adherence to HBV treatment”.

LINE 61: I do not totally agree with this statement: “Factors that contribute to non-adherence to medication stem from psychosocial, environmental, and health system influences” Side effects are one of the main reasons of suboptimal adherence and the authors do not mention it at any time.

LINE 66: “….with one study showing very low levels of effective antiviral therapy [9]” . check this, I do not understand what you mean.

LINE 106: “…from clinical and community-based organizations”; explain please what do you consider a community-based organization.

LINE 118: (e.g., CBOs) please, add the meaning of the acronyms at least the first time.

LINE 121: “After participants completed written informed consent approved by the Western Institutional Review Board (WIRB)” this does not go here.

TABLE 1: I do not like the way the information is shown as in the p-value column the figures are not aligned with the row.

TABLE 4: “Lived in the US >= 10 years (< 10 years)” check it, I don’t understand if it is more or less with those punctuation marks

LINE 284: rewrite the author Contribution’s section according to MDPI editorial policies and delete: “All the listed authors made significant contributions to the article”.

LINE 292: delete the Data Availability Statement and the template text:  “In this section, please provide details…if the study did not report any data” .

LINES 302-315: delete it, it is repeated from the template.

The Acknowledgment’s statement does not acknowledge anything so, if you want to include it please, mention your founders (or whoever), and if not remove it.

The statement: I would delete this: “Its contents are solely the responsibility of the authors and do not necessarily represent the official views of the NCI/NIH” but if it is finally included I think it should be in Institutional Review Board Statement or in Funding.

Kind regards.

The reviewer,

Author Response

(The authors gave the same response as above.)

Round 2

Reviewer 2 Report

You have corrected the mistakes and the article has improved so I would accept in present form.

Congratulations for your effort.

The reviewer.